# A New Perspective to Understand the Late Season Abundance of *Delia antiqua* (Diptera: Anthomyiidae): A Modeling Approach for the Hot Summer Effect

**DOI:** 10.3390/insects14100816

**Published:** 2023-10-16

**Authors:** Yong Kyun Shin, Subin Kim, Chung Gyoo Park, Dong-Soon Kim

**Affiliations:** 1Majors in Plant Resource Sciences & Environment, College of Applied Life Sciences (SARI), Jeju National University, Jeju 63243, Republic of Korea; skh0184@jejunu.ac.kr (Y.K.S.); tnqls643@jejunu.ac.kr (S.K.); 2Insect-Verse Laboratory, Jinju-daero 859-1, Jinju 52716, Republic of Korea; insectpark1@daum.net; 3The Research Institute for Subtropical Agriculture and Biotechnology, Jeju National University, Jeju 63243, Republic of Korea

**Keywords:** phenology, hot temperature, *Allium* pests, summer diapause, forecasting model

## Abstract

**Simple Summary:**

The onion maggot, *Delia antiqua*, is an important pest of crops belonging to the *Allium* genus worldwide. It is known that high temperature has a significant effect on the development of this pest, but how temperature conditions affect population abundances in the field environment has not been fully understood. Our model, which incorporated the hot summer effect into the model of summer diapause termination, showed that high temperatures in summer clearly delayed or suppressed the late season occurrence. This phenomenon was observed largely in the Jinju and Jeju regions of Korea, while it was not obvious in regions with cooler summers such as the United States, Canada, and Germany, where the occurrence peaks were overlapped or slightly separated between the generations before and after summer. Thus, our results provide a new perspective from which to understand the late season abundance of *D. antiqua*.

**Abstract:**

The onion maggot, *Delia antiqua* (Meigen), is one of the most important insect pests to agricultural crops within *Allium* genus, such as onions and garlic, worldwide. This study was conducted to understand the seasonal abundance of this pest, with special reference to the hot summer effect (HSE), which was incorporated into the model of summer diapause termination (SDT). We assumed that hot summer temperatures arrested the development of pupae during summer diapause. The estimated SDT curve showed that it occurred below a high-temperature limit of 22.1 °C and peaked at 16 °C. Accordingly, HSE resulted in delaying the late season fly abundance after summer, namely impacting the third generation. In Jinju, South Korea, the activity of *D. antiqua* was observed to cease for more than two months in the hot summer and this pattern was well described by model outputs. In the warmer Jeju Island region, Korea, the late season emergence was predicted to be greatly delayed, and *D. antiqua* did not exhibit a specific peak in the late season in the field. The abundance patterns observed in Korea were very different from those in countries such as the United States, Canada, and Germany. These regions are located at a much higher latitude (42° N to 53° N) than Korea (33° N to 35° N), and their HSE was less intense, showing overlapped or slightly separated second and third generation peaks. Consequently, our modeling approach for the summer diapause termination effectively explained the abundance patterns of *D. antiqua* in the late season. Also, the model will be useful for determining spray timing for emerging adults in late summer as onion and garlic are sown in the autumn in Korea.

## 1. Introduction

The onion maggot, *Delia antiqua* (Meigen) (Diptera: Anthomyiidae), is one of the most important insect pests to agricultural crops within the *Allium* genus: onion and shallot (*A. cepa* L.), leek (*A. ameloparsum* L.), garlic (*A. sativum* L.), and chives (*A. schoenoprasum* L.) [1,2]. This dipteran is found worldwide mainly in northern temperate regions throughout Europe, Asia, and North America [1,3], and has even been reported in Chile, South America [4]. *D. antiqua* adults lay eggs on or near the base of *Allium* plants, and the hatched larvae move into the soil and feed on the belowground parts of the plants (roots and bulb), sometimes entering through the basal stem [1,5]. The larvae have three instars to pupate in the soil at depths of 5–10 cm [5]. In autumn, cool temperatures and short day lengths induce winter diapause in this fly, i.e., a 50% induction at 14 °C under 16L:8D or 18.5 °C under 12L:12D [6], allowing it to overwinter in the soil, and emerge as adults in spring the next year [7].

*Delia antiqua* is a multivoltine pest species with a varying number of generations per year depending upon climatic environments. In northern temperature regions in the United States (i.e., New York), this pest has three generations annually during the growing season [7]: late May–late June, mid-July–mid-August, and late August through September [5]. Also, three generations were found to occur in Ontario, Canada [8,9], and in Mainz, Papenburg, and Bürstadt regions in Germany [10]. Thus, trivoltine is common in these temperate regions, although various unverified generation numbers by field data have been reported in Russia: monovoltine in regions of the far north (Murmansk, Igarka), two generations in Ukraine, and three to four generations in southern Kazakhstan in southern areas under favorable conditions, reviewed in [11].

The damage of *D. antiqua* is economically important in the production of onion and garlic worldwide. An attack of *D. antiqua* larvae is very destructive to *Allium* crops because a single larva can destroy an average of 28 young seedling onions (diameter 1 mm at soil surface, height 40 mm) over its development [12]. The damage could reach 57.4% to nearly 100% in New York [13,14,15] if is not treated with insecticides such as chlorpyrifos during planting. In Chile, the damage to onion seedlings reached 17.9% and was reported to be 35.8% on shallot [4]. Also, this fly can infest garlic with a maximum of 32.3 larvae/bulb in the Czech Republic [16] and has been found to cause serious damage in Korea, reviewed in [17]. 

According to Werling [18], the onion damage caused by *D. antiqua* larvae was high (37%) in the earliest planting (16 April) and lower (21%) in intermediate plantings (6 May) but economically unacceptable and lowest (2%) in late plantings (28 May). Since the first generation causes the greatest economic loss as above [13,18], forecasting models for predicting the spring emergence have been developed to support the spray timing, targeting the first generation in the United States [7], Canada [9], and Norway [19]. However, the circumstances may be different in Korea from the countries above, because onions and garlic are planted in autumn when the third generation of *D. antiqua* occurs. In Korea, most onions are grown from seed planted from late August through mid-November and are harvested in April to early June of the next year. Also, two types of garlic cropping systems are generally practiced: warm-season garlic and cold-season garlic. The former is a type that is sown in late August to early September and harvested in early May of the following year, while the latter is sown in late September to October and harvested in mid-to-late June of the following year. Consequently, information on the emergence time and abundance patterns of *D. antiqua* in the late season after summer is essential to preventing larval infestation during early growth stages in onions and garlic plantings.

Temperature has been recognized as one of the most important factors related to the damage and abundance of *D. antiqua* [20,21]. This may be largely linked to the biology of *D. antiqua,* which likes cool temperatures as reported by Moretti [21]: “The developmental optimum for *D. antiqua* has been estimated at approximately 22 °C [22,23], adult activity tends to decline in hotter temperatures [1], and oviposition ceases at temperatures exceeding 30 °C [24]”. Also, *D. antiqua* undergoes a summer diapause (i.e., aestivation), in which pupal development is sometimes arrested temporarily in high summer temperatures [25,26]. Because hot summers are not suitable for this pest [20], it seems to be repeatedly cited that the distribution of this fly is restricted mostly to the temperate zone of the Holarctic region (35–60° N) [1,20]. 

The hot summer effect (HSE), which is defined as the arrest of pupal development above a certain critical temperature and is closely linked to the maintenance of summer diapause (aestivation), can significantly affect the population abundance of *D. antiqua* in the late season after summer. This is because aestivation occurs between the second and third generations as described above [5,9,10]. Accordingly, we hypothesized that the larger the HSE, the more delayed the third generation, as seen in Figure 1. A strong HSE may have a devastating effect on the fate of *D*. *antiqua*. In the present study, we established a model that describes the termination of summer diapause; through this process, we incorporated a hot temperature effect into it and tested the adult emergence time from pupal cohorts in summer diapause comparing with various field data in Korea (Jinju, Naju and Jeju). Also, we discussed how the delayed occurrence affects the fate of *D. antiqua*, especially under global warming.

## 2. Materials and Methods

### 2.1. The Summer Diapause Termination and Post-Diapause Development

We constructed a model to describe the emergence of *D. antiqua* adults in summer diapause as the following two phases (Figure 2): The first phase indicates the process in which summer-diapausing pupae terminate diapause and move physiologically to a normal state. The state of normal pupae is defined as a pupal state in which the physiological age is set to 0 and can start development to emerge into an adult as soon as environmental conditions are allowed. The second phase involves the transformation of regular pupae (i.e., non-diapausing pupae) into adults, which is referred to as adult emergence. Thus, the former (Phase I) and latter (Phase II) become the summer diapause termination (SDT) model and stage transition model of pupae to adults, respectively.

In *D. antiqua*, the state of summer diapause is broken by a specific stimulus. The key factor was known as temperature, as reported by Ishikawa et al. [26]. They showed a temperature-dependent termination rate of summer diapause reaching a maximum rate at 16 °C. Thus, we assumed temperature primarily affected the SDT for the purpose of modeling. Also, post-diapause development, expressed by the stage transition model of pupae, was a temperature-dependent process as with poikilothermal organisms [27].

We used an equation-fitting professional software, TableCurve 2D [28], to estimate the parameters of model equations in all cases below.

#### 2.1.1. Model for the Summer Diapause Termination (SDT), Phase I

This process consisted of two unit models: the completion model of SDT and the distribution model of summer diapause completion time. The former determined the time of summer diapause termination by accumulating daily the development rate for summer diapause termination (hereafter, SDT development rate) according to temperature, while the latter arranged the probability distribution of summer diapause completion time from 0% to 100%.

***Completion model of SDT*.** The process of SDT in *D. antiqua* differed depending on the temperature to which the pupae were exposed [26]. In the report [26], the completed proportion (*y*) of SDT in a pupal cohort was a function of the exposure period (*x*) at a certain temperature. We assumed a linear relationship between *x* and *y*; accordingly, Equation (1) was applied without the intercept (i.e., *y* = 0 at *x* = 0).
(1)$$f\left(x\right)=ax,$$
where parameter *a* is the slope of the straight line, which is changed by treated temperatures [26]. This equation was used to interpolate a 50% completion time of summer diapause in the pupal population; except for 2 °C, where extrapolation was conducted (see Appendix A). For the estimation of parameters, we applied the data sets published previously by Ishikawa et al. [26]. They reported excellent data sets for our purpose: SDT rates obtained after 1, 2, 3, and 5 d exposures at 2.0, 5.6, 10.0, 15.8, and 21.0 °C, respectively.

Finally, the period [*P_50_*(*Ti*)] required for the 50% completion at temperature *Ti* could be defined as Equation (2).
(2)$$P_{50}\left(T_{i}\right)=\frac{1}{2a_{Ti}}\;,$$
where *a_Ti_* is the slope of the regression line at temperature *T_i_*. The reciprocal of *P_50_*(*T_i_*) was regarded as the SDT development rate of *D. antiqua* pupae, and we used the Lactin equation [29] to fit the data points of 1/*P_50_*(*T_i_*) as a function of temperatures (Equation (3)). This equation has been used as the development model for various insect species [30,31].
(3)$$r_{t}\left(T\right)=\left[\exp\left(\rho T\right)-exp\left(\rho T_{m}-\frac{T_{m}-T}{∆T}\right)\;\right],$$
where if *r_t_*(*T*) < 0 then *r_t_*(*T*) = 0, *r_t_*(*T*) is the SDT development rate at temperature *T* (°C). According to the definition of Logan et al. [32] and Damos and Savopoulou-Soultani [33], *T_m_* is the thermal maximum, Δ*T* is the temperature range over which “thermal breakdown” became the overriding influence, and *ρ* is the composite value for critical enzyme-catalyzed biochemical reactions in insect development. 

***Distribution model of summer diapause completion time*.** Insect species held under the same environments develop at different rates due to inherent differences among individuals [34]. Such individual variation occurs also in the SDT development as reported by Ishikawa et al. [26]. Also, since *P_50_*(*T_i_*) varies with temperature, a standardized distribution that is independent of temperature is required so that the data points converge to the center.

Each period (days) of the temperature exposure was normalized by dividing it by *P_50_*(*T_i_*) at each temperature, which was defined as normalized time. Finally, the data pairs of normalized times (x) and SDT rates (y) were fitted by the two-parameter Weibull function [34,35,36]; using the data sets published by Ishikawa et al. [26].
(4)$$f_{s}\left(x\right)=1-exp\;(-{\left(x/\alpha )\right)}^{\beta },$$
where *f_s_(x*) is the proportion of individuals who completed SDT at normalized time *x*, and *α* and *β* are parameters. 

***SDT density curve***. The SDT density curves concerning the days (i.e., the period of temperature treatment) and temperature were calculated using Equation (4), *f_s_*(*x_i_*). The normalized times were obtained by the equation below.
(5)$$x_{i}=\int_{t_{0}}^{t_{n}}r_{t}\left[T\left(t\right)\right]dt\approx \sum_{i=1}^{n}r_{t}(T_{i})∆t,$$
where temperature (*T*) is a function of time (*t*) and *r_t_*(*T_i_*) is the SDT development rate at temperature *T* (°C) of the *i*-th day. The Δ*t* is the time interval and is set to 1 because one-day intervals are applied. Thus, the proportion of pupal cohort in summer diapause shifted to the next stage (i.e., normal pupae) during the time interval between *i* and *i + 1* was calculated by subtracting the outputs of *f*(*x_i_*) from *f*(*x_i + 1_*).

#### 2.1.2. Stage Transition of Non-Diapausing Pupae, Phase II

The stage transition model simulates the proportion of individuals, which completed the development from non-diapausing pupae to adults. We basically used the modeling protocol developed by Kim et al. [37] for the simulation, which originated from Curry et al. [38,39] and Wagner et al. [40]. This process requires two unit models: the development model and the distribution model of development time [40].

***Development model*.** The data sets of temperature-dependent development for *D. antiqua* pupae (non-diapausing pupae) were obtained from Eckenrode et al. [7], Park [25], and Ishikawa et al. [26]. Data sets of Eckenrode et al. [7] and Ishikawa et al. [26] were digitalized by WebPlotDigitizer [41]: (constant temperature/development time in days), Eckenrode et al. [7] = 7.2/62.6, 10.0/39.9, 12.8/27.1, 15.6/17.0, 18.3/15.0, 21.1/11.0, 23.9/8.0 and 26.7/7.0; Ishikawa et al. [26] = 10.0/50.9, 15.8/22.0, 21.2/14.0 and 25.0/11.5. Also, Park’s data [25] were 10.0/38.3, 15.0/18.7, 20.0/11.6, 25.0/8.6, and 30/6.4 (see Appendix A). 

The reciprocal of the development time of pupae was used as the development rate to fit Equation (3) for the development model of pupae [*r_p_*(*T*)], as a function of temperatures.

***Distribution model***. The distribution model of development time in insect species has been constructed based on individual variation in the completion time of the development. Since individual variation data for the pupal development time were not available, we estimated the model parameters using the standard deviations and mean development times as suggested by Kim and Kim [42]. The frequency distribution was prepared using the data sets of Park [25] as provided in Appendix A). The normalized cumulative frequency distributions of pupal development time at each temperature were combined and fitted to Equation (4) for the distribution model of pupal development time [*f_p_(x*)] as a function of the independent variable, namely the physiological age of pupae. The physiological age of pupae (*px_i_*) was obtained by accumulating the *r_p_*(*T*) as shown in Equation (5). 

***Density curve.*** The same approach as the SDT density curves above was applied to simulate the daily emergence of adults in relation to temperatures and days. The development rates [by using *r_p_*(*T*)] were accumulated according to corresponding temperatures and used as input values for the distribution model *f_p_(x*) that arranged the emergence proportions through time.

### 2.2. Estimation of Thermal Constant (Degree Days) for Stage Development

The thermal constant (degree days, DD) was established to evaluate the stage development of *D. antiqua* after adult emergence from summer diapause pupae. Thermal constants for each developmental stage in DD were estimated using previously reported temperature-dependent development of *D. antiqua* (source data sets are available in Appendix A). We applied the concept of common lower threshold temperature (*T_cb_*) to facilitate DD calculation in practice, which uses a single base temperature for all stages [43]. To obtain *T_cb_* = 3.9 °C, we averaged the lower threshold temperatures of all stages: eggs = 3.1 °C [10], larvae = 3.8 °C [10] and 4.3 °C [6], and pupae = 4.0 °C [10] and 4.4 °C [7].

Total degree days (DD) (i.e., thermal constant *K*) required to complete development at each stage were calculated using the following Equation (6) [44]: (6)K=di (Ti−Tb),
where *T_i_* was the temperature *i* incubated, *d_i_* was the mean number of days in incubation at the *i*-th temperature, and *T_cb_* was the common lower threshold temperature (see Appendix A).

### 2.3. Model Simulation and Field Comparison

#### 2.3.1. Model Overview and Simulation

Our model describing the adult emergence of *D. antiqua* from summer-diapausing pupae consisted of two phases (I & II) with three developmental stages (or state) as seen in Figure 2: pupae in summer diapause (aestivating pupae), non-diapausing pupae, and adults. The model starts at the aestivating pupal stage of a single cohort (1000 pupae) with the assumption that the physiological state is the same (i.e., physiological age = 0). Each stage included daily-based multiple cohorts of individuals that entered the stage on a given day and were treated as different age groups in the stage [45,46]. Thus, our models operate in discrete time steps, although the equations of unit models have continuous attributes. Also, the stages move by the deterministic rule, which predicts the mean value of the process that considers the average effects as parametric values. However, the distribution models realized the stochastic aspect of the arriving mean value, which explicitly considered statistical variations.

At any given time, each daily cohort is characterized by two state variables as the same methods of Shaffer and Gold [47] and Kim and Lee [46]: *x_ij_*(*t*) = the physiological age of cohort j within stage *i* at time *t* and *N_ij_*(*t*, *x*) = the number of individuals in the cohort which are of physiological age *x* at time *t*. The output of the model is *N_i_*(*t*), the total number in stage *i* at a time *t*, which is obtained simply by summing of all the cohorts. Thus, the computations of the model are updated using the results in a daily time step (24 h).

A daily-basis sequential computational process was carried out for the simulation using the Microsoft Excel program (Appendix A). The model was operated from the first day of August since this is the hottest season. 

#### 2.3.2. Field Data Collection

The seasonal population abundance of *D. antiqua* was investigated in onions and garlic fields in 2021 (late March to mid-October) and 2022 (early March to late December) in Jeju, Korea, where approximately 1000 ha of onions and garlic are cultivated. *D. antiqua* adults are attracted significantly to white color [48]. Thus, a commercial white sticky plate (10 × 25 cm; Russell IPM, England) was rolled into a cylindrical shape (height 10 cm, diameter 7.6 cm) and used to monitor *D. antiqua* adults. The cover sheet on the inner surface was not removed to use only the outer surface of the cylindrical sticky traps. The traps were installed vertically at a height of 1 m using steel stakes. In 2021, two monitoring sites (33°15′02.57″ N/126°14′38.05″ E, and 33°15′29.83″ N/126°14′42.84″ E; ≈ 1 km apart from each other; each HSL (height above sea level) 33 m and 34 m, respectively) were selected and three traps were installed, with 5 m between traps at each site. An additional monitoring site (33°15′17.52″ N, 126°13′53.42″ E; HSL 40 m) was made in 2022 (thus additional three traps). The traps were replaced weekly and brought to the laboratory to examine the adults under the microscope (10× to 40×). The identification of *D. antiqua* adults was made using the morphological keys provided by Savage et al. [49]: hind tibia with 7–15 short erect posteroventral bristles, frontal vitta visible at the narrowest point of frons, and parafacial broad in lateral view. The trap data were combined each year to examine the abundance patterns. In addition, in 2022, the occurrence of immature stages of *D. antiqua* after the summer diapause was investigated in garlic fields in Jeju, where traps were installed.

Other field data sets for the adult flight activity were obtained from the previous reports [50,51] in various regions in Korea (Appendix A). These field data sets were beneficial for comparing the population abundances of *D. antiqua* among the climatically different environments. 

Additionally, we obtained the cohort emergence data from Park et al. [51] in 1986 and 1987 in Jinju, in which pupae entering summer diapause were treated in the soil (a 5 cm depth) and the adult emergence was examined in autumn (Appendix A).

#### 2.3.3. Meteorological Data

Weather data (daily minimum and maximum air temperatures) were obtained from the Korea Meteorological Administration (KMA). We used corresponding region weather data nearest where the field phenology was monitored (see Appendix A). The air temperatures (*T_air_*) were converted to soil temperatures (*T_soil_*) using the equation developed by Choi et al. [52].
(7)$$T_{Soil}=\left\{\begin{array}{c} T_{air}+0.9,\;\;\;if\;T_{ave\_air}<23.3\;\\ {\;T}_{air}+0.6,\;\;\;\;if\;T_{ave\_air}\geq 23.3\;\; \end{array}\right\}.$$

This equation used two parameters of daily average air temperatures (*T_ave_air_*) and a reference temperature (23.3 °C), which was empirically developed based on the actual observation measured at a depth of 5 cm in soil at six sites in Jeju. Temperature at a depth of 5 cm in soil was frequently used in emergence experiments of *D. antiqua* adults in Korea [17,53], since 90% of pupae were found within a depth of 10 cm in the soil [54].

#### 2.3.4. Model Comparison with Field Data

The model was evaluated by comparing the outputs with field data. First, we examined how the outputs of the model were positioned in the seasonal abundance of *D. antiqua* adults in different regions. Then, the cohort emergence data were used to explore the model accuracy in the prediction timing.

To examine the population development in the late season, tracking of the stage development of *D. antiqua* was carried out at the 10% adult emergence time that emerged from the pupal cohorts in summer diapause (*n* = 1000). Using the common lower threshold temperature, the development until pupae (434 DD) and the completion of the egg-egg period (651 DD) were tracked. 

In 2022 in Jeju, fortunately, we collected larvae (*n* = 50) on 24 November, and brought them to the laboratory to rear into pupae (Temperature 23 °C, a photoperiod of 16L:8D). The larvae pupated within 24 h following the collection so that it was confirmed that they were at least in the late state of the larval stage, namely “1 d = 19.1 DD” was required for pupation when *T_cb_* = 3.9 °C was applied. This information was used to track backward the presence of *D. antiqua* adults (Appendix A). So, 414.8 DD was applied for tracking backward (433.9 DD–19.1 DD, here 433.9 DD indicates thermal requirement until the completion of larval stage).

The cohort emergence data in 1986 (*n* = 154) and 1987 (*n* = 155) in Jinju [51] were each accumulated and scaled to the total number for the purpose of comparison with the model outputs. The model outputs were also accumulated and scaled to the maximum value (i.e., total number). For the statistical test, data pairs (Julian dates) were obtained by extracting paired data points from the actual and model at 10, 15, 25, 50, 75, 90, and 95% cumulative emergence. To find exact data pairs at each percentage as possible, data points were interpolated between two successive observation dates, and then a one-sample *t*-test was applied for a null hypothesis that the mean discrepancy in days between observed and predicted dates equals 5 days [55]. 

## 3. Results

### 3.1. The Summer Diapause Termination and Post-Diapause Development

#### 3.1.1. Model for the Summer Diapause Termination (SDT)

***Development model of SDT.*** The development rates of SDT were well described with the Lactin equation as indicated by a high determination coefficient (Figure 3A, R^2^ = 0.97), and the regression model was statistically significant (*F* = 38.19; *df* = 2, 2; *p* = 0.0255). The curve peaked at approximately around 16.0 °C with a high-temperature limit of 22.1 °C (Parameter, the thermal maximum). The estimated parameter values are seen in Table 1.

***Distribution model of SDT time***. The variation of the SDT time (summer diapause completion time) was well fitted to the Weibull function, as described in Figure 3B. The relative proportion for the SDT time was transitionally changed along normalized time. The estimated parameter values of the distribution function are shown in Table 1. The equation fitting was statistically significant (*F* = 838.05; *df* = 1, 23; *p* < 0.0001; R^2^ = 0.97). At the approximate normalized time 1.1546, the completion rate of SDT became 63.2% (i.e., parameter α). 

***SDT density curve***. The SDT density curves concerning the days and temperature showed a gentle hill-like peak (Figure 3C). At low temperatures, the ridge of the completion rate was low and wide. It became gradually higher and narrower to form the peak as temperature increased. After the peak, the elevation decreased sharply and dropped to a low level with a long range at high temperatures. 

#### 3.1.2. Stage Transition of Non-Diapausing Pupae

***Development model.*** The nonlinear relationship between the development rate and temperature for pupae of *D. antiqua* fitted well with the Lactin equation as seen in Figure 4A. The development rate increased almost linearly up to 30.0 °C and dropped sharply to 37.9 °C (parameter *T_m_*). The regression analysis was statistically significant (*F* = 105.19; *df* = 2, 14; *p* < 0.0001; R^2^ = 0.94), and the estimated parameters are provided in Table 2.

***Distribution model***. The Weibull function was fitted satisfactorily to the variation of pupal development time as seen in Figure 4B. The transition occurred in a somewhat wide range at the center (parameter *α* = the transition center and *β* = the steepness, Table 2). The regression was highly significant statistically (*F* = 3935.97; *df* = 1, 43; *p* < 0.0001; R^2^ = 0.99).

***Density curve.*** The predicted daily number of individuals that emerged from the pupal stage to adults in relation to cohort age and temperature are shown in Figure 4C. Density curves showed a shape similar to that of the ridge of a mountain range gently enveloping the surroundings. At low temperatures, the transition density curves had a larger meantime and higher variation (i.e., a long transition period). As the temperature increased, the mean transition time and variance decreased, showing sharper peaks. 

### 3.2. Estimation of Degree Days for Generation Time

The degree days (i.e., thermal constant) for the development completion of each stage of *D. antiqua* based on a common lower threshold temperature (*T_cb_* = 3.9 °C) were calculated as 53.6 DD for eggs, 79.3 DD for the first instar, 73.6 DD for the second instar, 159.2 DD for the third instar, 217.0 DD for pupae, and 68.2 DD for the pre-oviposition period of females (Appendix A). Consequently, the period for the egg to egg of *D. antiqua* was estimated to be 650.9 DD.

### 3.3. Model Simulation and Comparison with Field Data

To compare the late season abundance patterns of *D. antiqua* in various regions with different climatic conditions, the model results were shown together with the actual data investigated in this study, including the previous data sets (Figure 5). The previous data for 1980 (Naju) and 1985 (Jinju) were insufficient to understand the occurrence pattern due to the small amount of trap catches, but the data sets for 1986 and 1987 in Jinju and 2021 and 2022 in Jeju showed sufficient occurrence to compare the occurrence pattern. In Jinju, the adult populations of *D. antiqua* were apparently discontinued for more than two months in the hot summer and showed large peaks in autumn, but the seasonal abundance pattern was largely different in Jeju. That is, the adult populations of *D. antiqua* did not form a specific peak abnormally in the late season after summer. Actually, we found only three adults on traps from November to December 2022 in Jeju. The outputs of the model that predicted the adult emergence time of the pupae in summer diapause were well positioned in the actual autumn observation period of *D. antiqua* in Jinju from 1985 to 1986. The emergence times predicted by the model were much delayed in Jeju, showing a delay of more than one month compared to those in Jinju.

In Jeju, the occurrence of later generations of *D. antiqua* was negligible, so it was impossible to validate the model outputs with actual field data. Instead, the backtracking result provided the model performance. The backward tracking from the collection of larvae (Julian date 330 = 24 November 2022) showed that *D. antiqua* adults should at least be present on 24 October 2022 (Julian date 298). This appearance of adults by backtracking was approximately in the middle of the daily emergence distribution period predicted by the model (please find the bent arrow in Figure 5F). 

The generation development of *D. antiqua* to the pupal stage after adult emergence was predicted in early to mid-October in Jinju and Gwangju regions when the development was tracked from the 10% adult emergence time of pupal cohorts in the summer diapause (see ✖ symbol in Figure 5). However, in the case of the Jeju region, it was possible to develop into a pupa in early November in 2022 and, in 2021, the pupa could emerge in early December when the weather was cold. In addition, in Jeju in 2021, the degree days required to complete the time of the egg-to-egg generation were not formed by the end of the year.

To validate the accuracy of the emergence model developed in this study, field-observed and model-predicted adult data were compared using the cohort emergence data sets in 1986 and 1987 (Figure 6). The model outputs successfully predicted the actual emergence curves in both years, although model outputs were slightly delayed compared to the actual observation. The discrepancy between the actual and predicted dates by the model was not different from 5 days in 1987, as tested by a one-sample *t*-test (Table 3), showing an overall difference in 4.8 days, but the difference increased to 7.6 days in 1986.

## 4. Discussion

### 4.1. Model Parameters and Simulation

High temperatures during the summer months arrest the pupal development of *D. antiqua* temporarily (aestivation) [25,26]. Unlike the summer dormancy in the congener species of *D. radicum*, where it is terminated instantaneously when the pupae are placed at <20 °C [56], *D. antiqua* requires a period of time to complete the SDT [10]. Ishikawa et al. [26] applied that the development of arrested pupae resumed if mean daily habitat temperatures dropped below 19.5 °C. In the present study, however, we applied directly the SDT development model without a critical temperature for the initiation of SDT completion, since the parameter of *T_m_* = 22.1 °C plays the role of the critical temperature. We also used a 3 d moving average temperature to stabilize the starting point of the model. Consequently, the hot summer effect was successfully incorporated into the model by parameterization for the relationship between temperature and SDT development rates and also temperature-dependent characteristics in SDT development were well reflected as seen in Figure 3A.

The accumulated SDT rates are served to run the distribution model of summer diapause termination time, which distributes stochastically the completion time of SDT over time. This logic is the same as that of the normal pupae stage transition model consisting of the pupal development model and distribution model of development time. The protocol was conceptualized and realized by Curry et al. [38,39], Wagner et al. [40], and Kim et al. [37], and was applied to project the phenology under variable field temperatures in various insect pests [30,31,57]. The pupae in the summer diapause arrived in the state of non-diapausing pupae as they completed the SDT, and then the daily cohorts of non-diapausing pupae underwent the development into adults. In this way, the termination model of the summer diapause was completed structurally with biological parameters and was successfully simulated. 

The model results could well explain the delaying occurrence patterns of *D. antiqua* in the late season, which was actually observed in Jinju, Korea, although there were some errors in the accuracy of the model outputs. In the simulation of the Jeju region, where the weather is warmer, the hot summer effect was clearly reflected, such as the delay of the later generations. In this way, we think that this model reflects the biological characteristics of this fly’s response to high temperatures well. Regarding the error in our model results, it may be caused by various external factors as well as the model parameter itself. That is, the discrepancy might be related to differences in some extent between weather station temperatures used to run the model and local temperature and soil temperature in the field. Soil temperature converted from air temperature seems to be applicable to the operation of our model because we used the equation that was developed based on the actual measured temperature data in the soil [52]. This approach may be effective because air temperatures are much easier and less costly to record, and soil temperature must be measured at many points by the heterogeneous profile of the soil [7]. In a similar approach, Choi et al. [58] converted air temperatures to soil temperatures by simply adding 0.8 °C, and this method was very effective in predicting the phenology of soil-inhabiting pests of *Agrotis ipsilon* in Korea. Despite the limitation for field validation our model, the model outputs showed considerable consistency with actual observation, especially in Jinju, Korea. As defined by Jeffers [59], any model is a simplified description of the full system it represents, and a model should express the key essence of the system within the model assumption. Our model captured well the key point in the late season abundance of *D. antiqua* caused by the hot summer. Since our model was developed by incorporating the innate biological characteristics of *D. antiqua*, unlike the empirical model, it can be further improved through field validation in the future as noted by Shaffer and Gold [47] and Kim and Lee [46]. Also, the model structure and making process presented in this study can be sufficiently used to develop a model for other insect pests that have a similar aestivation ecology. 

### 4.2. Comparison of Model Outputs with Field Data and Its Application

The patterns of population abundance of *D. antiqua* observed in Jinju and Jeju in Korea were very different from those in Elba, Potter, and Oswego in New York, United States [7], Keswick Marsh in Ontario, Canada [9], and Mainz in Rheinland-Pfalz, Papenburg in Niedersachsen and Bürstadt in Hessen, Germany [10]. In these countries, the second and third peak periods appeared to be somewhat overlapped or slightly separated from each other. This pattern may be explained by the first assumption as presented in Figure 1A: a weak hot summer effect. As seen in Appendix A, there was no visible separation between the two last peaks in Mainz and Elba, and a weak separation was found in Papenbrug and Potter (a delay of about 20 days or so), where mean air temperatures somewhat exceeded the critical temperature of 22.1 °C. In Elba, the temperature change was similar to that of the Porter area, but the high-temperature effect was not evident. This is probably the result of a complex interaction between the population size (see the maximum peak size, i.e., approximately 400 adults in Elba: Appendix A) and the intensity of the summer-diapause induction stimulus. In other words, according to the study of Ishikawa [26], it is known that the critical temperature for the summer-diapause induction (SDI) of *D. antiqua* is approximately 24 °C regardless of the photoperiod. In these areas, it is judged that the SDI stimulus appeared intermittently (i.e., over 24 °C). Therefore, it is presumed that the effect of high-temperature in summer is weak because only some individuals are subjected to SDI stimulus, and when the population size is large, the individuals not subjected to SDI are more dominant, resulting in a weak hot summer effect as shown in Elba. However, the situation can be very different in Korea, which is at a much lower latitude than these regions (Appendix A) and is affected by constant high temperatures in summer (Figure 5).

The abundance pattern of *D. antiqua* in Jinju, Korea (Figure 5B,C) well represents the second assumption with a moderately hot summer effect, in which a complete third peak appears with a much-delayed form. This kind of pattern is also found in La Platina in the north-central area of Chile, where there was a lull of more than a month for the activity of *D. antiqua* adults between summer and autumn [4]. La Platina (S33°34′ S) of Chile is located in the southern hemisphere and it may be understandable for the abundance pattern of *D. antiqua* there in that it is at a similar distance in terms of latitude, which is opposite to Jinju (N35°1′) in Korea (see Appendix A). Chile is not recorded yet as a distribution area of *D. antiqua* in CABI or EPPO Global Database of plant pest distribution. Although Gomes et al. [60] suggested that the records of the *Delia* species in South America require review because they may be *D. sanctijacobi*, but the distribution of *D. antiqua* in Chile seems possible from a climatic point view (Appendix A). This hypothesis will need to be further tested in the future. 

The occurrence pattern in the Jeju area is the first observed phenomenon that has not been found anywhere else in the world (Figure 5E,F). In the model, the adult emergence in the late season (i.e., the third peak) was much delayed, and was not actually detected by trap monitoring. We observed that only small quantities of adults occurred in the late season as examined by backward tracking from the date of larvae found in the field. This finding was consistent with assumption C with a strong hot summer effect in Figure 1. A similar penology was reported in the carrot root fly, *Chamaepsila rosae* (Fabricius), a root-feeding maggot. The adult flight of the ‘third’ generation after summers with pronounced hot periods was extremely weak in the fields [61]. In France, furthermore, the second generation after summer did not occur in very hot years [62]. It was understood that the delayed occurrence mediated by pupal aestivation linked to hot weather increased the mortality of *C. rosae*, which led to failure in the late season [2]. In the occurrence pattern of *D. antiqua* in Jeju, the delayed period binds this fly to be subjected to pressure by various biotic and abiotic mortality factors [1,2], and if this operates, it can explain the weak late abundance. Also, one possibility is that winter-diapause ecology may be involved. Winter diapause in *D. antiqua* is known to be induced in the pupal stage [63] and is mainly dependent on the combined effects of photoperiod and temperature [6,63,64]. It is doubtful whether pupae that have finished summer diapause can go directly into winter diapause by external stimulant cues, and this needs to be examined in the future. Otto and Hommes [10] modeled the induction of winter diapause by which the pupae will enter diapause if the mean daily soil temperature drops below 13.5 °C over 3 days or longer, because of no available information. 

Taken all together, accordingly, the geographical distribution of *D. antiqua* may be determined primarily by the hot summer effect related to the aestivation biology. So, hot temperatures due to global warming are likely to have a devastating effect on the late-season population of *D. antiqua*, as in Jeju, where the late season occurrence was insignificant, and the climate is subtropical. Perhaps hte Jeju area may lie at one of the borders of the distribution of *D. antiqua* since the occurrence of this fly is unconfirmed in many regions that are warmer than Jeju. With further warming, this fly will disappear in the Jeju area as in most tropical areas. There is one case in which *D. antiqua* has been reported in areas with tropical climates: that is, the report for the incidence and control of this fly at some irrigation sites in the Upper East Region of Ghana in Africa [65], where temperature conditions are not suitable for *D. antiqua* (Appendix A). As suggested by Gomes et al. [60], the case in Ghana could be a temporary occurrence after the misidentification of another species or an accidental introduction, and it is judged that continuous settlement is impossible according to our model. As such, although our hypothesis will require further testing in many regions of the world, we were able to explain the key phenomena of *D. antiqua* abundance patterns through our model. In any case, the current form of our model can be applied usefully in practice.

In Korea, the cropping system of warm-season garlic is common in relatively warm areas such as Jeju and Jinju. Therefore, a forecasting model for adult emergence in the late season is urgently required to determine the optimal time for adult control after sowing onions and garlic. Our model will be useful for this purpose until more advanced models are developed. The management of *D. antiqua* has relied on prophylactic insecticide applications at planting [13,15,66,67]. In recent years, spraying insecticides against adult flies has become an option to control *D. antiqua,* as conventional soil insecticide use in agricultural fields is limited to groundwater protection issues, although insecticide seed treatment is a crucial option [68,69]. Furthermore, the termination model of summer diapause for *D. antiqua* developed in the present study can significantly improve previous or new population models since the previous population model [10] poorly predicted the third generation after summer due to the failure of incorporating the module for aestivation biology.

## Figures and Tables

**Figure 1 insects-14-00816-f001:**
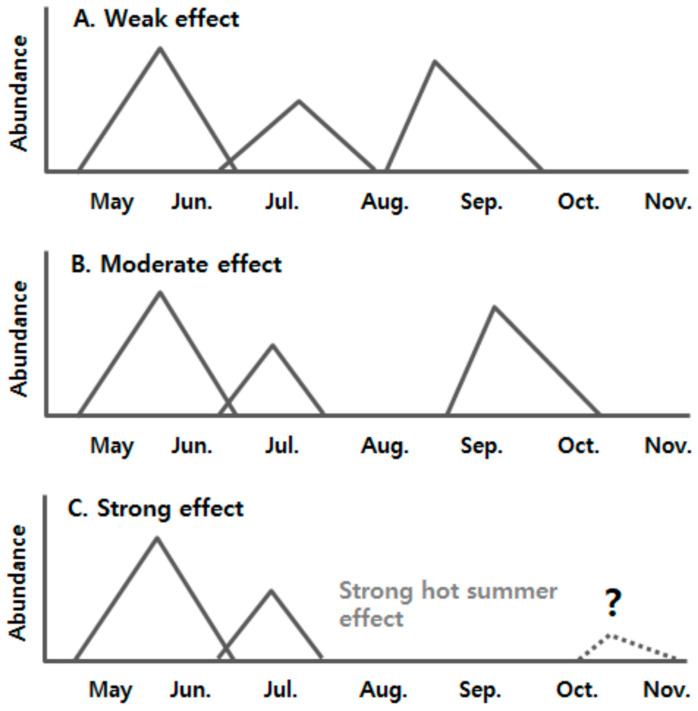
Hypothesized seasonal abundance patterns of *D. antiqua* by hot summer effect linked to the summer diapause behavior (i.e., aestivation). We hypothesized that the larger the hot summer effect, the more delayed the third generation.

**Figure 2 insects-14-00816-f002:**
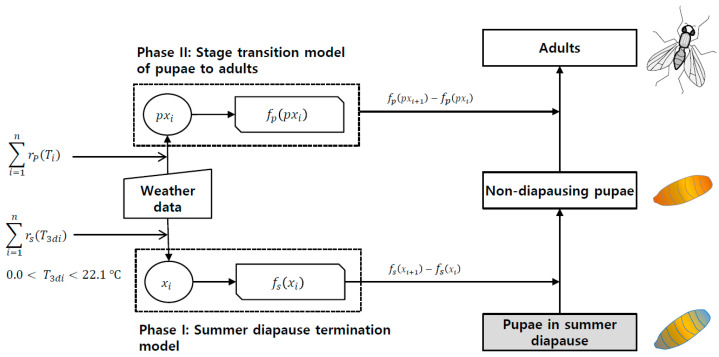
Schematic diagram for the simulation process of the summer diapause termination and the emergence of pupae to adults in *D. antiqua* population. $r_{s}(T_{3di})$ = completion rate model of summer-diapause termination, $r_{p}(T_{i})$ = development rate model (1/days) of non-diapausing pupae, $f_{P}\left({px}_{i}\right)\;$= distribution model of pupal development time, ${px}_{i}$ = physiological age of pupae at *i*-th day, $f_{s}\left(x_{i}\right)$ = distribution model for the completion time of summer-diapause termination, $x_{i}$ = physiological age of pupae in summer-diapause at *i*-th day, and $T_{3di}$ = three-day moving average temperature.

**Figure 3 insects-14-00816-f003:**
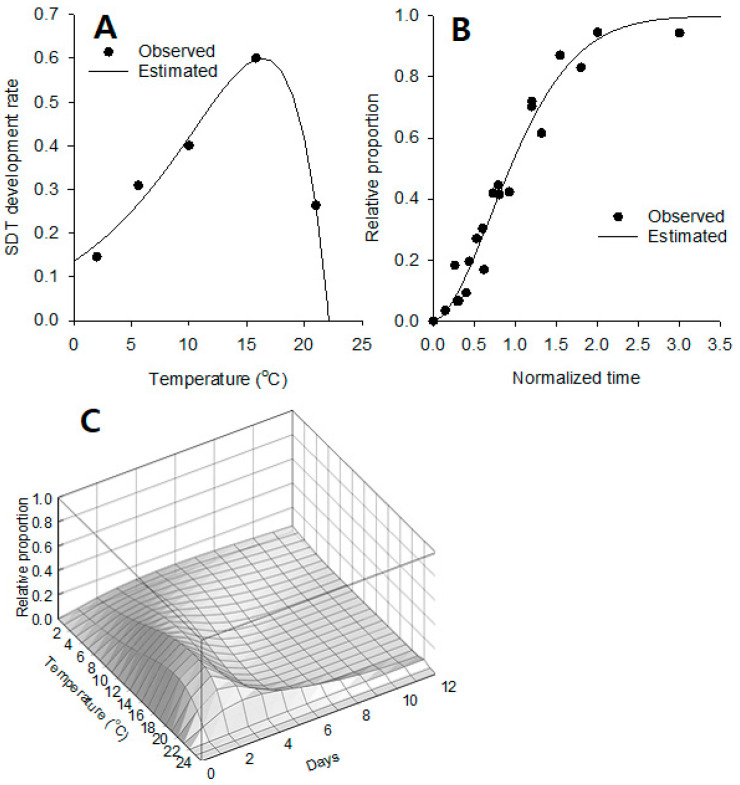
Models describing the summer-diapause termination (SDT) in *D. antiqua*. The relationship between SDT development rate and temperature (**A**), distribution model of summer-diapause completion time by normalized time (**B**), and SDT density curves in relation to the days and temperature (**C**). The observed data sets were obtained from the published data of Ishikawa et al. [26].

**Figure 4 insects-14-00816-f004:**
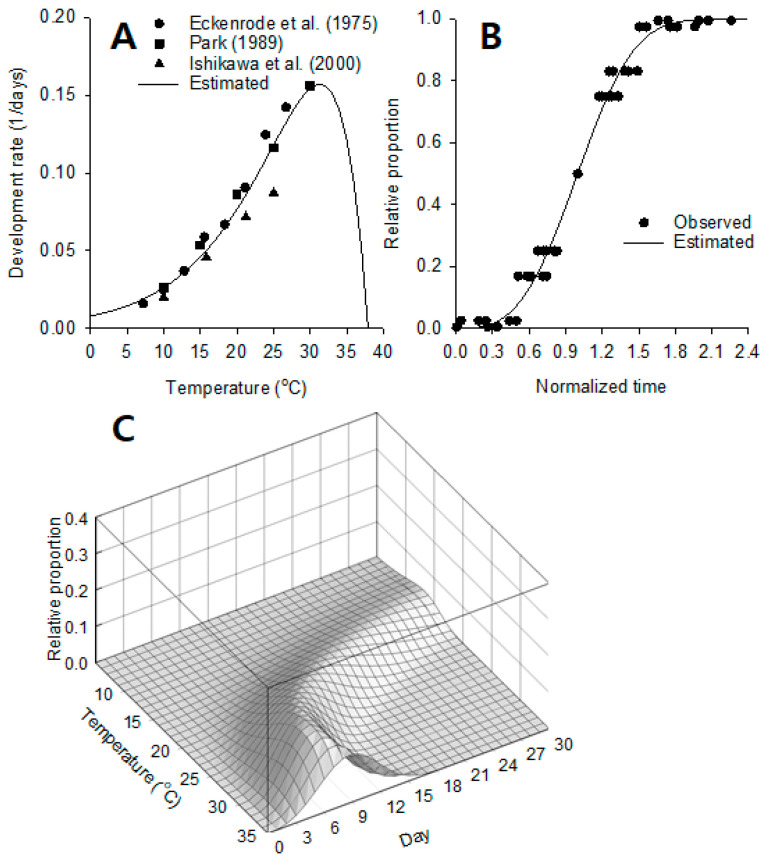
Component models for the stage transition of non-diapausing pupae in *D. antiqua*. Development rate (1/mean development time in days) as a function of temperature (**A**), distribution model of development time (**B**), and predicted density curves of pupal stage transition in relation to age (day) and temperature (**C**) [7,25,26].

**Figure 5 insects-14-00816-f005:**
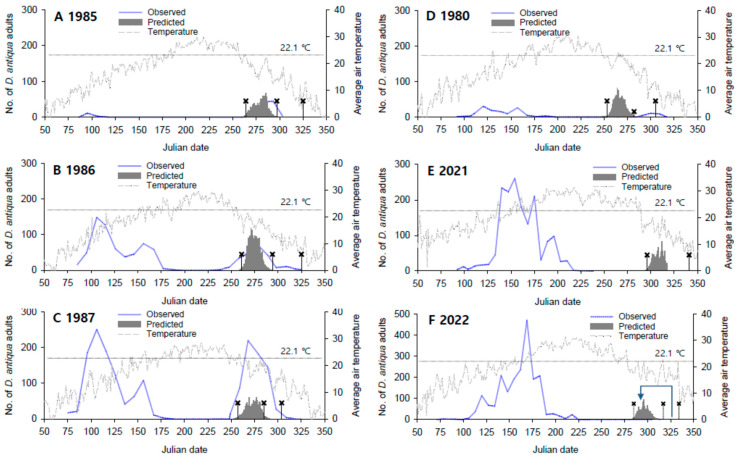
The predicted adults emergence of *D. antiqua* from summer diapausing cohorts (*n* = 1000) by the simulation model, comparing with the seasonal abundances. (**A**–**C**) in Jinju [51], (**D**) [50] in Naju and (**E**,**F**) in Jeju, Korea. The stage development is noted by the ✖ symbol: the first, second and third ✖ indicate the starting date of degree-day calculation at 10% adult emergence, the development until pupae (433.9 DD), and the completion of the egg–egg period (650.9 DD), respectively. The bent arrow on Figure (**F**) means backward tracking to the adult emergence from the late larval stage. The horizontal dotted lines indicate the critical temperature of 22.1 °C for SDT development.

**Figure 6 insects-14-00816-f006:**
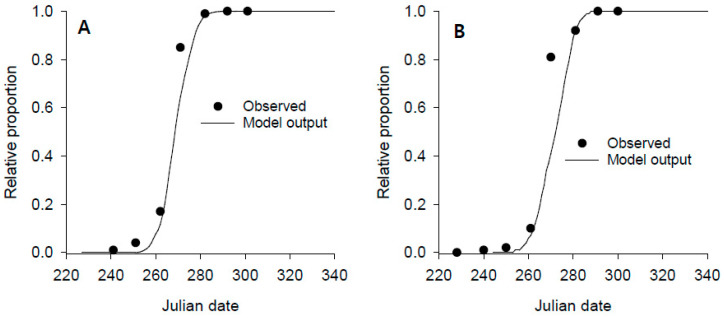
Comparison of model outputs with actual observed data for predicting adult *D. antiqua* emergence. (**A**,**B**) are accumulated adult emergence curves from pupal cohorts in summer diapause in 1986 (*n* = 154) and 1987 (*n* = 155), respectively.

**Table 1 insects-14-00816-t001:** Estimated parameter values of the component models describing the summer diapause termination of *D. antiqua* pupae.

Model	Parameter	Estimated	SE	R^2^
Termination of summer diapause	ρ	0.16978	0.013258	0.97
	$T_{m}$	22.13802	0.268383	
	∆T	5.67016	0.395566	
Distribution of summer diapause completion	α	1.15460	0.036266	0.97
	β	1.71717	0.119902	

**Table 2 insects-14-00816-t002:** Estimated parameter values of the component models describing the stage transition of *D. antiqua* pupae: the development rate curve and distribution of development time of pupae.

Model	Parameter	Estimated	SE	R^2^
Development rate	ρ	0.151932200	0.019070840	0.94
	$T_{m}$	37.88922166	3.322196970	
	∆T	6.572996371	0.822062604	
Distribution of development time	α	1.121146325	0.010050286	0.99
	β	3.155020615	0.121914537	

**Table 3 insects-14-00816-t003:** Accuracy of days in forecasting the date (Julian) of each percentage of cumulative emergence of *D. antiqua* adults from pupal cohorts in summer diapause.

Percent Emergence	1986			1987		
	Observed	Predicted	Difference ^a^	Observed	Predicted	Difference
10	256.0	267.0	11.0	261.0	265.0	4.0
15	260.0	268.0	8.0	261.5	266.5	5.0
25	263.5	270.0	6.5	263.5	268.5	5.0
50	266.5	273.5	7.0	267.5	274.5	7.0
75	269.5	278.5	9.0	269.5	279.0	9.5
90	275.0	282.0	7.0	279.0	282.0	3.0
95	279.0	284.0	5.0	284.0	284.0	0.0
Mean			7.64 *^b^			4.79 ns
SE			0.7296			1.331
*t*-value			3.501			0.1891

^a^ Difference between observed and predicted date in absolute value. ^b^ One-sample *t*-test was applied for a null hypothesis that the mean discrepancy in days between observed and predicted dates at 10, 15, 25, 50, 75, 90, and 95% cumulative emergence equals 5 days: ns not significant; * *p* < 0.05.

## Data Availability

The data that are presented in this study are available in the article and Supplemental information.

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
