# Peer review of "A New Perspective to Understand the Late Season Abundance of Delia antiqua (Diptera: Anthomyiidae): A Modeling Approach for the Hot Summer Effect"

_insects, 2023, doi:10.3390/insects14100816_

Round 1
Reviewer 1 Report
Manuscript ID: insects-2617312 titled “A new perspective to understand the late season abundance of Delia antiqua (Diptera: Anthomyiidae): a modeling approach for the hot summer effect” by Yong Kyun Shin, Su-bin Kim, Chung Gyoo Park, and Dong-Soon Kim, submitted to section: Insect Ecology, Diversity and Conservation is a well written article with loaded information on , theoretical population biology ecology comparing the population dynamics of D. antiqua among the climatically different environments in Korea. Forecasting models for predicting the spring emergence have been developed to support the spray timing targeting the first generation of this pest. I support consideration for publication in INSECTS pending minor revision. The attached PDF has some notes for authors revision. Some suggested notes are:
· Please advise if this submission to insects after withdrawal from Journal of Crop Protection. A preprint of the same manuscript available at: file:///C:/Users/ramadanmm/Downloads/SSRN-id4394529.pdf
· What is the developmental stage inside the puparium during aestivation, developed pupa (with head, thorax, abdomen, inside the puparium), or larval prepupal form inside the puparium. Please clarify.
· References in text are not in sequence, please rearrange.

Author Response
General response: Thank you for the valuable comments. We respect the comments and suggestions, and have revised our manuscript accordingly. Most of them were corrected directly in the revised manuscript. For comments that need our response, we have provided our response with point-by-point to the comments. The changes are shown in Track Change mode or red color.
Specific response to the comments
Line 29, add (average temp. ): In Jinju, Korea, the activity of D. antiqua was observed to cease for more than two months in the hot summer and this pattern was well described by model outputs.
Response: Thank you for the comment.
The sentence above is closely related to the initiation of SDT (summer diapause termination). So, we add about the SDT response to temperature as below,
The estimated SDT curve showed that it occurred below a high-temperature limit of 22.1 °C and peaked at 16 °C.
Line 33, add average elevations
Response: Thank you for the comment.
The meaning of this sentence was the location of the sites. We have changed the sentence as following;
These regions are located at a much higher latitude (42° N to 53° N) than Korea (33° N to 35° N), and their HSE was less intense showing overlapped or slightly separated the second and third generation peaks.
Line 49, add (average temp)
Response: Thank you for the comment.
The sentence was modified as followings;
In autumn, cool temperatures and short day lengths induce winter diapause in this fly, i.e., 50% induction at 14 ℃ under 16L:8D and 18.5℃ under 12L:12D [6]
Line 66, ( add name of insecticide used )
Response: Thank you for the comment.
We added representative chemical: “such as chlorpyrifos”
Line 95, reference sequence ?; Line 157, references not in sequence Line 159, references not in sequence
Response: Thank you for the comment.
The order of reference was totally updated as the sequence.
Line 271, add elevation (m) on sites
Response: Thank you for the comment.
The elevation based on HSL were provided as below,
(33°15'02.57"N/126°14'38.05" E, and 33°15'29.83"N/126°14'42.84" E; 1 km apart from each other; each HSL (height above sea level) 33 m and 34 m, respectively)
(33°15’17.52˝N, 126°13’53.42˝E; HSL 40 m)
Line 295, Ref 46?
Response: Thank you for the comment.
The error was corrected as [52]
Line 432, Table 3
Response: Thank you for the comment.
The error of 277.0 at 90% in the column of ‘observed (1986)’ was corrected to 275.0
Reviewer 2 Report
This paper presents a model that incorporates high temperature related summer diapause, the “hot summer effect,” to explain late season abundance of the onion maggot. They show that high summer temperatures clearly delay or suppress late season occurrence in the lower latitudes of Korea. Their model provides a remarkably good fit for populations of flies in Korea. I have no concerns or criticisms beyond noting that there are a few English language issues. These issues can be addressed with editing to improve the grammar.
See above
Author Response
Thank you for the positive evaluation of our manuscript. We checked again to improve English grammar. The changes will be shown in Track Change mode.
Reviewer 3 Report
A new perspective to understand the late season abundance of Delia antiqua (Diptera: Anthomyiidae): a modeling approach for the hot summer effect
In the present study, a model explains the summer diapause termination of Delia antiqua, a major onion pest. Based on my reading, the manuscript is well-written, well-structured, and clear. The methods are effectively presented, and the results are characterized in a way that is likely to be interesting to readers. Overall, I enjoyed it. However, I have a few minor suggestions below:
1- Line 12: Please change to “belonging to the Allium genus worldwide”.
2- Line 46: D. antiqua adults lay eggs
3- Line 84: Figure 1. I am interested to know if this figure is included in your results. I am uncertain about its appropriate placement.
4- Line 116: please remove “from pupae”
5- Line 118” Please remove “the summer”
6- Line 119: Please remove “of pupae”
7- Line 121 “The second phase involves the transformation of regular pupae into adults, which is referred to as adult emergence.”
8- Line 125: "I concur with your viewpoint, although I believe that the photoperiod may also play a role."
9- Lines 241-242: Not clear, please provide more clarification.
10- Line 243 and elsewhere: summer diapausing pupae (aestivating pupae), non-diapausing pupae???
11- Lines 241-242: my question is, " After diapause termination, summer diapausing pupae undergo metamorphosis and either transform into non-diapausing pupae or directly develop into the adult stage.?
12- Line 265 and 450: D. antiqua
13- Line 466: Please note that diapause is not an abnormality. Therefore, I recommend using non-diapausing pupae instead of normal pupae
As far as I know, the manuscript is written in fluent English and only requires minor editing.
Author Response
General response: Thank you for the valuable comments and positive evaluation. We respect the comments and suggestions, and have revised our manuscript accordingly.
We have provided our response with point-by-point to the comments. The changes are shown in Track Change mode.
Specific response to the comments
1. Line 12: Please change to “belonging to the Allium genus worldwide”.
Response: Thank you for the comment.
We have corrected as recommended.
2. Line 46: D. antiqua adults lay eggs
Response: Thank you for the comment.
We have corrected as recommended.
3. Line 84: Figure 1. I am interested to know if this figure is included in your results. I am uncertain about its appropriate placement.
Response: Thank you for the comment.
Figure 1 was arranged in the front to illustrate the hypotheses of this study. By doing this, we thought readers would be able to see the overall structure of the paper and gain greater understanding.
Figure 1 was useful in organizing the entire study in the discussion section rather than the results. I would be very grateful if you would allow me to leave it where it is.
4. Line 116: please remove “from pupae”
Response: Thank you for the comment.
We have removed the phrase of “from pupae”
5. Line 118” Please remove “the summer”
Response: Thank you for the comment.
We have removed the phrase of “the summer”
6. Line 119: Please remove “of pupae”
Response: Thank you for the comment.
We have removed the phrase of “of pupae”
7. Line 121 “The second phase involves the transformation of regular pupae into adults, which is referred to as adult emergence.”
Response: Thank you for the comment.
The sentence was modified as suggested.
8. Line 125: "I concur with your viewpoint, although I believe that the photoperiod may also play a role.“
Response: Thank you for the comment.
We have not yet found that photoperiod is involved in the termination of summ diapause in this species as well as similar species.
9. Lines 241-242: Not clear, please provide more clarification.
“Our model describing the adult emergence of D. antiqua from summer-diapausing pupae consisted of summer diapause termination and pupal stage emergence with three developmental stages (or state) (Figure 2): pupae in summer diapause, normal pupae, and adults”
Response: Thank you for the comment.
The sentence was changed as follows; “Our model describing the adult emergence of D. antiqua from summer-diapausing pupae consisted of two phases (I & II) with three developmental stages as seen in Figure 2: pupae in summer diapause, non-diapausing pupae, and adults”
10. Line 243 and elsewhere: summer diapausing pupae (aestivating pupae), non-diapausing pupae ???
Response: Thank you for the comment.
We used preferly the term “non-diapausing pupae” throughout the manuscript.
11. Lines 241-242: my question is, " After diapause termination, summer diapausing pupae undergo metamorphosis and either transform into non-diapausing pupae or directly develop into the adult stage.?
Response: Thank you for the comment.
As the response of # 9, he readability of the sentence was improved.
12. Line 265 and 450: D. antiqua
Response: Thank you for the comment.
We have corrected as recommended.
13. Line 466: Please note that diapause is not an abnormality. Therefore, I recommend using non-diapausing pupae instead of normal pupae.
Response: Thank you for the comment.
We used preferly the term “non-diapausing pupae” throughout the manuscript as that in # 10.